# Enhancing Generative Retrieval with Reinforcement Learning from Relevance Feedback

**Yujia Zhou[1], Zhicheng Dou[2]\*, Ji-Rong Wen[2]**

[1]School of Information, Renmin University of China
[2]Gaoling School of Artificial Intelligence, Renmin University of China
{zhouyujia,dou,jrwen}@ruc.edu.cn

## Abstract

The recent advent of end-to-end generative retrieval marks a significant shift in document retrieval methods, leveraging differentiable search indexes to directly produce relevant document identifiers (docids) in response to a specific query. Nevertheless, this approach faces two fundamental challenges: (i) a discrepancy between the token-level probabilistic optimization and the broader document-level relevance estimation; (ii) an overemphasis on top-1 results at the expense of overall ranking quality. To tackle these challenges, we propose a generative retrieval model with reinforcement learning from relevance feedback, which aims to align token-level docid generation with document-level relevance estimation. The training process incorporates three stages: supervised fine-tuning, relevance reward model training, and reinforced learning-to-rank from relevance feedback. To train a high-quality reward model, we define "relevance" under three progressive scenarios, which collectively offer a comprehensive evaluation of the document relevance. Experiments conducted on two benchmark datasets demonstrate the effectiveness of our proposed approach.

## 1 Introduction

Information retrieval systems have undergone progressive advancements over the past few decades (Zhu et al., 2023). Historically, the retrieval of candidate documents has been heavily reliant on the use of an inverted index (Robertson and Zaragoza, 2009), a conventional method that employs term-based parsing. With the emergency of neural networks, dense retrieval (Karpukhin et al., 2020; Xiong et al., 2021) is proposed to encode both query and document into dense vectors, so as to measure the similarity in semantic space.

Recently, there has been a growing focus on investigating end-to-end generative retrieval models (Tay et al., 2022; Bevilacqua et al., 2022; Wang

et al., 2022; Zhou et al., 2022a). These models supersede the conventional explicit index by implementing a large-scale model known as the differentiable search index (Tay et al., 2022). This innovative approach facilitates end-to-end document retrieval using a seq-to-seq generative model, such as T5 (Raffel et al., 2020). In this framework, the model directly generates the corresponding docid for a given query.

Despite the notable progress made in generative retrieval models, they still face two major challenges that limit their ability to retrieve relevant documents. First, these models primarily rely on the auto-regressive loss to maximize the token-level generation probability of the ground-truth docid. However, such an optimization goal is not aligned with the primary objective of ranking tasks, which is to estimate document-level relevance. Second, existing approaches assume a deterministic distribution in which all probability mass is allocated to the "golden" docid. Such an assumption can lead the model to overemphasize the top-1 generation results, while ignoring the quality of the overall ranking list. In light of these challenges, establishing alignment between generation probabilities and document relevance is a vital necessity in adapting generative models to ranking tasks.

In this paper, we present a Generative Retrieval model with Reinforcement Learning from relevance feedback (GenRRL). Our objective is to align the generative retrieval model, trained specifically on the task of docid generation, with the model responsible for assessing document relevance. The training process comprises three pivotal steps: (1) Supervised fine-tuning. This initial step involves training the model using labeled data to improve its performance in generating docid. (2) Relevance reward model training. The subsequent step focuses on training a relevance reward model, specifically designed to measure the degree of relevance between the generated docids and their correspond-

---

*Corresponding author

ing queries. We expect the reward model to assign higher scores to docids that are more relevant to the queries. (3) Reinforced learning-to-rank (L2R) from relevance feedback. The final step involves using the trained relevance reward model to guide the generative retrieval system towards generating more relevant docids. To align with the ranking task requirements, we incorporate the idea of L2R into the reinforcement learning procedure. This enables the provision of not only pointwise reward but also pairwise and listwise relevance feedback, thereby enhancing the overall ranking performance.

In order to train a high-quality relevance reward model, it is crucial to define "relevance" accurately within the context of document retrieval tasks. In this regard, we focus on three key scenarios that robustly reflect the relationship between queries and documents. Firstly, term-based overlap concerns the presence of identical or similar words within both the query and the document. Techniques like TF-IDF can effectively gauge this overlap. Secondly, semantic similarity extends beyond mere term overlap, delving into the shared meanings or concepts within the query and the document. Dual encoder models are particularly useful in modeling the semantic similarity by encoding the query and the document into dense vectors. Lastly, contextual dependency considers how effectively the document satisfies the information needs expressed in the query, extending beyond the boundaries of specific terms or concepts within the query. Techniques for capturing contextual dependency could include language models, which have demonstrated proficiency in understanding the contextual relationships between two sentences. By considering these three progressive scenarios, we can comprehensively assess the relevance between queries and documents, leading to a more effective reward model.

To assess the effectiveness of our model, we perform comprehensive experiments on widely adopted MS MARCO and NQ document retrieval datasets. Experimental results demonstrate the efficacy of the proposed method, including the diverse relevance annotations and the reinforcement learning process. Our in-depth analysis on different model sizes also reveals the scaling laws of our proposed method.

Our contributions are summarized as: (1) We employ reinforcement learning from relevance feedback to improve generative retrieval, aiming to align the docid generation process with document relevance assessment. (2) We outline a reinforced training process specifically tailored for the ranking task, which includes relevance reward model training and reinforced learning-to-rank. (3) We provide a classification of "relevance" within the context of document retrieval. By considering three progressive aspects, the trained reward model enables a thorough assessment of the relevance between queries and documents.

## 2 Related Work

Traditionally, index-based document retrieval methods encompass both sparse and dense retrieval techniques. Sparse retrieval methods (Dai and Callan, 2019, 2020; Bai et al., 2020; Formal et al., 2021), such as BM25 (Robertson and Zaragoza, 2009), utilize inverted indexes to assess term importance and calculate matching scores between queries and documents. Dense retrieval methods have been employed to overcome the limitations of word mismatching (Gao et al., 2021). These methods convert queries and documents into dense vectors and employ ANNS (Jégou et al., 2011) for efficient vector-based search. The dual encoder approach incorporating pre-trained language models has shown improved performance in dense retrieval. Techniques like hard negative sampling (Xiong et al., 2021; Karpukhin et al., 2020; Zhan et al., 2021; Guu et al., 2020) further enhance effectiveness.

In recent times, generative retrieval has gained significant popularity in the field of information retrieval (IR) tasks. A model-based IR approach, where docids are embedded into the model, was first introduced in (Metzler et al., 2021; Tay et al., 2022). Since then, several other models (Wang et al., 2022; Zhuang et al., 2022; Zhou et al., 2022b) incorporating a query augmentation module have been developed to enhance the performance. Moreover, the model-based IR framework has been extended to knowledge-intensive language tasks, leading to notable improvements in performance (Chen et al., 2022a; Bevilacqua et al., 2022; Chen et al., 2022b). Additionally, a series of studies by Sun et al. (2023); Chen et al. (2023); Ren et al. (2023); Li et al. (2023) explored various methods for representing docids to retain docid semantics.

In this paper, we focus on aligning the optimizing objective of generative retrieval models with document-level relevance, so as to alleviate the bias against the objective of ranking tasks.

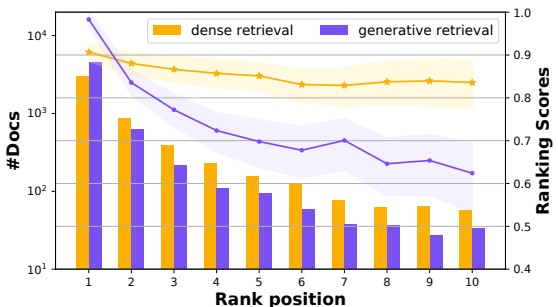

Figure 1: Empirical study on ranking results of generative retrieval and dense retrieval.

## 3 Preliminary

In this section, we formulate the generative retrieval problem and investigate its limitations.

### 3.1 Task Definition

Generative retrieval aims to generate the relevant docids for a given query directly with a seq-to-seq model. Assuming that $d'$ is the identifier of the document $d$, the generative retriever takes the query $q$ as input and generates the relevant docid with an auto-regressive score, written as:

$$\text{score}(d, q) = G_\phi(d'|q) = \prod_{i=1} G_\phi(d'_i|d'_{<i}, q), \quad (1)$$

where $G_\phi$ is the generative retriever with parameters $\phi$, and $d'_i$ is the $i_{th}$ token of the docid. The parameters $\phi$ is optimized with the standard seq-to-seq objective. During inference, the model applies constrained beam search to guide the decoder in searching within a limited token space at each step, so as to generate valid docids from a candidate set.

### 3.2 Empirical Study

In this section, we present an empirical study aiming to investigate the limitations of generative retrieval models compared to advanced dense retrieval models in ranking tasks. We choose DPR (Karpukhin et al., 2020) and Ultron (Zhou et al., 2022b) as representatives of the dense and generative retrieval model respectively, and investigate their ranking results on NQ dataset.

As shown in Figure 1, it can observed that: Firstly, the generative retrieval model demonstrates a higher likelihood of placing the correct document at the top position, while the dense retrieval model exhibits a more balanced distribution across all rank positions. Secondly, the relevant documents positioned at the first rank receive remarkably high

scores from the generative model, whereas the ranking score curve of the dense retrieval model appears smoother. This outcome is likely due to the discrepancy between the generation probability and the document relevance. To enhance the model ability on document relevance estimation, we propose a reinforcement learning framework for generative retrieval as follows.

## 4 Methodology

In this section, we focus on the training pipeline of GenRRL, including supervised fine-tuning, relevance reward model training, and reinforced L2R from relevance feedback.

### 4.1 Supervised Fine-tuning

Supervised fine-tuning (SFT) entails training the model to generate relevant docids based on token-level generation probabilities for a given query. Specifically, we choose keyword-based docids (e.g. URL, summary) as the training target, as these can more intuitively reflect the semantics of the document. In addition, we select standard T5 as the pre-trained model and apply pseudo query generation for data augmentation, which has been proven to be effective in improving the performance of generative retrieval (Zhuang et al., 2022). Formally, the generative loss function can be formulated as the maximization of the likelihood of the target sequence using teacher forcing, we have:

$$\begin{aligned} \mathcal{L}_{\text{gen}} &= -\log G_\phi(d'|q) \\ &= -\sum_{i=1} \log G_\phi(d'_i|d'_{<i}, q), \end{aligned} \quad (2)$$

where $G_\phi(d'_i|d'_{<i}, q)$ is the generation probability of token $d'_i$ based on the given input. After supervised fine-tuning, our next goal is to train a reward model designed to incorporate relevance preferences into docid generation process.

### 4.2 Relevance Reward Model Training

A high-quality relevance reward model is capable of effectively assessing the relevance between queries and documents. To accomplish this, we conduct an analysis of relevance judgments within the document retrieval. Subsequently, we employ three distinct methodologies to annotate document relevance, thereby facilitating the training of a comprehensive relevance feedback model. The details are introduced as follows.

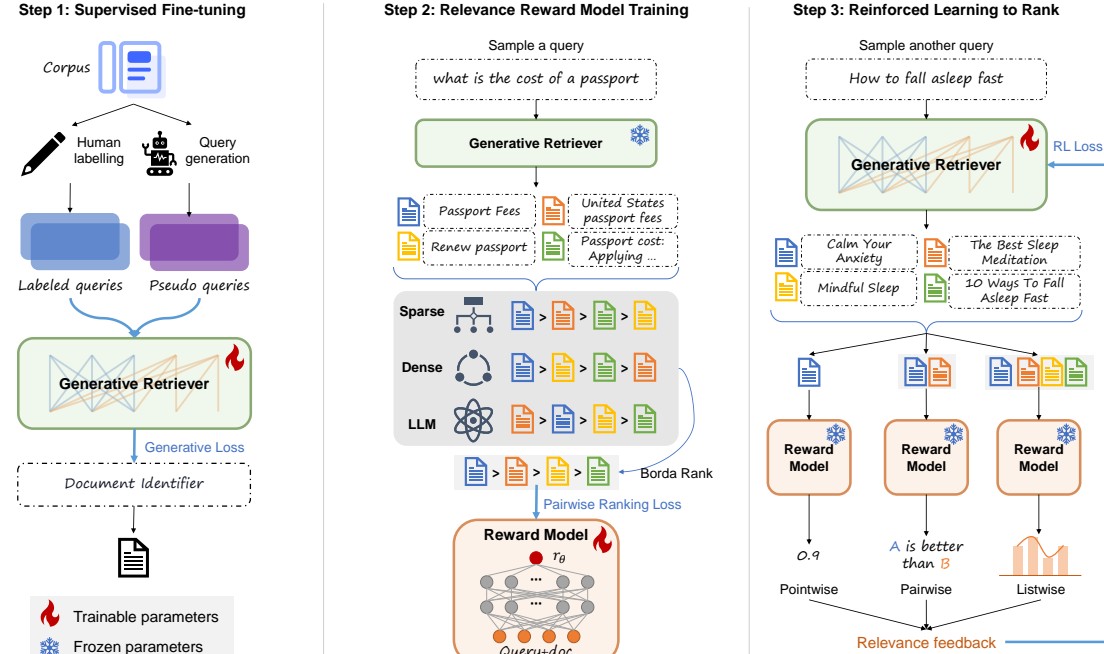

Figure 2: The training pipeline of GenRRL with supervised fine-tuning, relevance reward model training, and reinforced learning-to-rank from relevance feedback. Blue arrows indicate data used for model training.

**Relevance Classification.** We examine three scenarios wherein queries and documents exhibit strong associations. They are:

- Term-based Overlap. This scenario refers to the presence of identical or similar terms in both the query and the document. If a document contains many of the same words that are in the query, it is likely to be relevant.
- Semantic Similarity. This goes beyond term overlap to look at the meaning of the words and phrases in the query and document. Even if a document does not contain the exact terms present in the query, it might still be relevant if it discusses the same concepts. For example, a query for "cardiovascular fitness exercises" might retrieve a document about "aerobic workouts" because the two phrases refer to similar concepts.
- Contextual Dependency. This scenario refers to the extent to which a document fulfills the information needs expressed by the query, surpassing the limitations posed by specific terms or concepts. For instance, if a user has been searching for "how to save battery of laptops". A relevant document might say: "reducing screen brightness and turning off keyboard backlight".

**Relevance Annotation.** For each sampled query $q$ from supervised fine-tuning dataset, we generate a set of candidate documents based on the fine-tuned model, which corresponds to its own beam search results specific to that query. In order to capture the relevance of the above three scenarios respectively, we use three different models to annotate the relevance of the sampled query-doc pairs: **(1) To measure the term-based overlap**, we leverage BM25 (Robertson and Zaragoza, 2009), a well-established ranking function used by search engines. BM25's strength lies in its ability to weigh query terms based on their TF-IDF importance, thus providing a robust measure of term-based overlap. **(2) To capture the semantic similarity**, we employ the dual encoder model DPR (Karpukhin et al., 2020), which works by separately encoding the query and document into dense vectors in a shared embedding space. It focuses on deeper semantic connections between words and phrases in the query and the document. The dot product of the two vectors then gives a measure of the semantic similarity. **(3) For assessing contextual dependency**, we employ the large language model (LLM), LLaMA-13b (Touvron et al., 2023), to compute the generation likelihood of the document given a specific query. We assume that the generation probability can serve as an indicative metric of how well the document fulfills the implicit demands enclosed in the query.

To mitigate the discrepancies arising from variations in scoring distributions across different meth-

ods, rankings are employed as a means of introducing regularized supervision signals. Specifically, when provided with multiple rankings of the candidate documents, we adopt a voting-based approach, Borda Rank, to consolidate them into a unified ranking. For example, given $N$ items and $M$ individual rankings, the Borda rank score $B_i$ for item $i$ is calculated as $B_i = \sum_{j=1}^{M} (N - \mathrm{rank}_j(i))$, where $\mathrm{rank}_j(i)$ represents the rank of item $i$ in the $j$-th individual ranking. Based on ranking results, we can construct document pairs including a positive document $d^+$ and a negative document $d^-$ for reward model training.

**Reward Model training.** To train our reward models, we start from the supervised fine-tuned model, and then add a randomly initialized linear head that outputs a scalar reward. The reward model is trained in a pairwise manner to predict which document is more relevant. Given the query $q$ and a document pair $(d^+, d^-)$ sampled from the ranking $D$, the loss function for the reward model is:

$$\mathcal{L}_{\mathrm{RM}} = \mathbb{E}_{(q,d^+,d^-)\sim D}[\log(\sigma(r_\theta(q,d^+) - r_\theta(q,d^-)))],$$

where $r_\theta(q,d)$ is the scalar output of the reward model with parameters $\theta$.

Now we have an initial model capable of docid generation and a preference model that assigns a score to assess the document relevance. Subsequently, reinforcement learning is employed to optimize the initial model based on the reward model.

### 4.3 Reinforced L2R from Relevance Feedback

To begin with, we cast the generative retrieval task as a reinforcement learning (RL) problem. Initially, the policy is defined as a language model that accepts a query as input and generates a sequence of tokens as output. The action space of the policy encompasses all tokens in the language model's vocabulary, while the observation space corresponds to the distribution of potential input queries.

**Reward Function.** To guide the RL training, the reward function is constructed by combining the rewards $r_\theta(q,d)$ generated by the reward model with a KL divergence constraint, which ensures that the policy does not deviate significantly from its initial behavior, defined as:

$$R(q,d) = r_\theta(q,d) - \beta \log \left[ \pi_\phi^{\mathrm{RL}}(d|q) / \pi^{\mathrm{SFT}}(d|q) \right],$$

where $\pi_\phi^{\mathrm{RL}}$ is the learned RL policy, $\pi^{\mathrm{SFT}}$ is the supervised fine-tuned model, $\beta$ is The KL reward coefficient controlling the strength of the KL penalty.

To align with the requirements of the ranking task, we introduce the concept of learning-to-rank into the RL procedure, employing pointwise, pairwise, and listwise optimization strategies.

**Pointwise Optimization.** We begin by independently updating the model for each query-document pair. For a given query $q$, the generative retriever functions as a policy network, generating a candidate docid corresponding to the document $d$. The RL loss can be formulated as:

$$\mathcal{L}_{\mathrm{RL}} = -(R(q,d) - b) \sum_t \log G_\phi(d'_t | d'_{<t}, q),$$

where $b$ is the baseline incorporated to mitigate variance during RL training. It is computed by averaging the rewards of all samples.

**Pairwise Optimization.** Instead of regarding each query-document pair independently, we incorporate the assessment of pairwise document comparisons as a form of reward. Given a query $q$, the docid generation model samples plenty of document pairs $(d_i, d_j)$. The generator model is trained with the pairwise loss, denoted as:

$$\mathcal{L}_{\mathrm{RL}} = - \sum_{(d_i, d_j)} (R(q,d_i) \log p_{ij} + R(q,d_j) \log p_{ji}),$$

where $p_{ij}$ is computed by $|G_\phi(d'_i|q) - G_\phi(d'_j|q)|$ with the normalization of logistic function.

**Listwise Optimization.** Listwise strategy provides an alternative approach by considering the entire list of generated documents. In this context, we define a listwise loss that aims to minimize the ranking discrepancy between the SFT model and the reward model. Given a set of sampled candidates $C$ for the query $q$, the loss function can be defined as:

$$\mathcal{L}_{\mathrm{RL}} = - \sum_{d \in C} R(q,d) \log \frac{\exp(G_\phi(d'|q))}{\sum_j \exp\left(G_\phi(d'_j|q)\right)}.$$

By incorporating listwise optimization, the generative retriever is promoted to generate docids that closely resemble the ordering preferred by the reward model. This approach takes into account the overall quality and relevance of the entire list of generated documents, rather than focusing solely on an individual document or comparisons.

## 5 Experimental Setup

### 5.1 Datasets and Evaluation Metrics

Our experiments are conducted on two widely used document retrieval benchmarks: MS MARCO Doc-

ument Ranking (Nguyen et al., 2016) and Natural Questions 320k (Kwiatkowski et al., 2019). MS MARCO Document Ranking dataset is a large-scale collection specifically designed for the document ranking task. We create a candidate document subset of approximately 320k articles based on the labeled documents. Natural Questions (NQ) dataset comprises actual questions along with corresponding Wikipedia articles. During the dataset processing, we extract the title, abstract, and content from each Wikipedia article while removing unnecessary HTML tags. We eliminate duplicate documents based on their URL, which remains around 231k articles, with 307k training pairs and 7.8k test queries. To assess the ranking performance of document retrieval, we report MRR and Hits (at cutoff 1, 5, 10) as the evaluation metrics.

## 5.2 Baselines

To establish baselines for comparison, we consider three classes of models: (1) Sparse Retrieval models. We select **BM25** (Robertson and Zaragoza, 2009), which incorporates tf-idf weighting to determine term importance, and **DocT5Query** (Nogueira et al., 2019), which expands a document with queries predicted by a T5 model, as baselines. (2) Dense Retrieval models. We consider two dual encoder models trained with different hard negatives sampling strategies **DPR** (Karpukhin et al., 2020) and **ANCE** (Xiong et al., 2021). (3) Generative Retrieval models: **DSI** (Tay et al., 2022), the pioneering generative retrieval model that uses a seq-to-seq model to map the query to the relevant docid. Building upon DSI, **SEAL** (Bevilacqua et al., 2022) uses n-grams within passages as potential identifiers. **DSI-QG** (Zhuang et al., 2022), **NCI** (Wang et al., 2022), and **Ultron** (Zhou et al., 2022b) incorporate query generation modules for data augmentation. To ensure a fair comparison, all baseline models employ the "base" version of pre-trained models.

## 5.3 Implementation Details

**Docid.** We explore two types of linguistic docids: document URL and document summary. For the summary-based docids, we employ the LLaMA-13b model to generate document summaries using the prompt "*Summarize the following paragraphs with meaningful keywords: {document}*".

**SFT.** Our SFT model utilizes the T5-base pre-trained model (Raffel et al., 2020) as the back-bone and sets the learning rate and the batch size as 1e-3 and 128 during training. To enrich the training data, we employ the existing DocT5Query model (Nogueira et al., 2019) to generate 10 pseudo queries for each document.

**RL.** The reward model is initialized using the SFT model, with a linear head that is randomly initialized. During the training of the reward model, we employ the SFT model to generate 8 candidate docids for relevance annotation. We set the batch size to 64 and utilize a learning rate of 3e-4 for this training phase. In the reinforced L2R process, the policy model provides 1, 2, 4 candidate docids at each step corresponding to pointwise, pairwise, and listwise optimization, while the learning rate is set to 3e-5. All experiments are conducted on up to 8 NVIDIA A100 GPUs.

## 6 Experimental Results and Analysis

In this section, we present the experimental findings of our proposed approach and perform an empirical analysis to provide comprehensive insights.

### 6.1 Overall Results

The overall results of models are listed in Table 1. Some findings are summarized as follows.

**Comparison with Generative Retrievers**. Our observations indicate consistent superior performance of GenRRL compared to baseline models and SFT models when evaluated on the MS MARCO and NQ datasets. This pattern of results serves as compelling evidence for the effectiveness of our reinforced training strategy. Notably, when compared to the leading generative retrieval model Ultron on the MS MARCO dataset, GenRRL (URL) exhibits a noteworthy improvement of over 9.6% in terms of MRR@10. Similarly, on the NQ dataset, ROGER (Sum) surpasses the competitive baseline NCI by over 8.1% in MRR@10. These findings strongly support the notion that integrating relevance feedback into the initial generative retrieval model holds immense promise for enhancing performance in ranking tasks.

**Effect of Different Docids**. In the context of generative retrieval, the selection of docids can be categorized into two forms: linguistic docids and ID-based docids. Through a comparative analysis between DSI-QG and Ultron, we have observed that linguistic docids are better suited for generative retrieval tasks. Building upon this insight, we have

Table 1: Overall results. Document Rep. indicates the method to represent document. The best results are shown in **bold**. "†" indicates the result is significantly better than the corresponding SFT model with paired t-test at p < 0.05.

| Model | Document Rep. | MS MARCO | | | | Natural Questions | | | |
|---|---|---|---|---|---|---|---|---|---|
| | | Hits@1 | Hits@5 | Hits@10 | MRR@10 | Hits@1 | Hits@5 | Hits@10 | MRR@10 |
| Index-based Retrieval | | | | | | | | | |
| BM25 | sparse terms | 18.94 | 42.82 | 55.07 | 29.24 | 14.06 | 36.91 | 47.93 | 23.60 |
| DocT5Query | sparse terms | 23.27 | 49.38 | 63.61 | 34.81 | 19.07 | 43.88 | 55.83 | 29.55 |
| DPR | dense vector | 29.08 | 62.75 | 73.13 | 43.41 | 22.78 | 53.44 | 68.58 | 35.92 |
| ANCE | dense vector | 29.65 | 63.43 | 74.28 | 44.09 | 24.54 | 54.21 | 69.08 | 36.88 |
| Generative Retrieval | | | | | | | | | |
| DSI | semantic id | 25.74 | 43.58 | 53.84 | 33.92 | 27.42 | 47.26 | 56.58 | 34.31 |
| DSI-QG | semantic id | 28.82 | 50.74 | 62.26 | 38.45 | 30.17 | 53.20 | 66.37 | 38.85 |
| SEAL | n-grams | 27.58 | 52.47 | 61.01 | 37.68 | 29.30 | 54.12 | 68.53 | 40.34 |
| NCI | semantic id | 29.54 | 57.99 | 67.28 | 40.46 | 32.69 | 55.82 | 69.20 | 42.84 |
| Ultron | title + URL | 29.82 | 60.39 | 68.31 | 42.53 | 33.78 | 54.20 | 67.05 | 42.51 |
| SFT (URL) | doc URL | 29.90 | 60.28 | 68.17 | 42.52 | 33.91 | 54.18 | 66.92 | 42.52 |
| SFT (Sum) | doc summary | 30.12 | 60.68 | 68.71 | 42.84 | 33.72 | 54.71 | 67.69 | 42.83 |
| GenRRL (URL) | doc URL | 33.01† | 63.62† | 74.91† | 45.93† | 35.79† | 56.49† | 70.96† | 45.73† |
| GenRRL (Sum) | doc summary | **33.23**† | **64.48**† | **75.80**† | **46.62**† | **36.32**† | **57.42**† | **71.49**† | **46.31**† |

Table 2: The results of different relevance annotators. ● and ○ indicates annotating with and without the model.

| Model | Relevance annotator | | | MS MARCO | | Natural Questions | |
|---|---|---|---|---|---|---|---|
| | Sparse | Dense | LLM | Hits@10 | MRR@10 | Hits@10 | MRR@10 |
| SFT Model (Sum) | ○ | ○ | ○ | 68.71 | 42.84 | 67.69 | 42.83 |
| GenRRL (Sum) | ● | ● | ○ | 74.37 | 45.85 | 70.40 | 45.32 |
| GenRRL (Sum) | ● | ○ | ● | 73.19 | 45.18 | 69.78 | 44.67 |
| GenRRL (Sum) | ○ | ● | ● | 75.36 | 46.28 | 70.92 | 45.81 |
| GenRRL (Sum) | ● | ● | ● | 75.80 | 46.62 | 71.49 | 46.31 |

explored two types of linguistic docids, namely doc URL and doc summary. The experimental results demonstrate that our model consistently outperforms purely supervised fine-tuned models on both types of linguistic docids. Furthermore, the performance is particularly enhanced when utilizing summary-based docids, indicating that docids generated by large language models can better capture document relevance compared to the more simplistic approach of URL-based docids.

## 6.2 Ablation Study on Relevance Annotators

In this section, we perform an ablation study to assess the impact of relevance annotation in the GenRRL model. As described in Section 4.2, the relevance annotation process involves three types of annotators: the sparse model, the dense model, and LLM, each representing distinct relevance scenarios. We explore various combinations of these annotators to evaluate their effectiveness.

Table 2 presents the results for variants of GenRRL. We can observe that each annotator in the relevance annotation process demonstrates a contri-

bution to the overall performance, with the dense model exhibiting the most notable improvement in results. This finding suggests that the dense model possesses a more pronounced advantage in assessing the relevance between queries and documents. Additionally, the inclusion of the LLM annotator yields performance enhancements, indicating that leveraging the zero-shot capability of large language models can further augment the model's ability to capture contextual dependencies.

## 6.3 Comparison of Different RL Losses

The incorporation of relevance feedback signals with reinforcement learning serves as a crucial step in model fine-tuning. To investigate the impact of different reinforced feedback strategies, we compare three reinforcement learning losses: pointwise, pairwise, and listwise, and examine their influence on ranking performance for both type of docids.

Table 3 presents the results obtained for both URL-based docids and summary-based docids. It is observed that as the loss transitions from pointwise to pairwise and listwise, there is a gradual im-

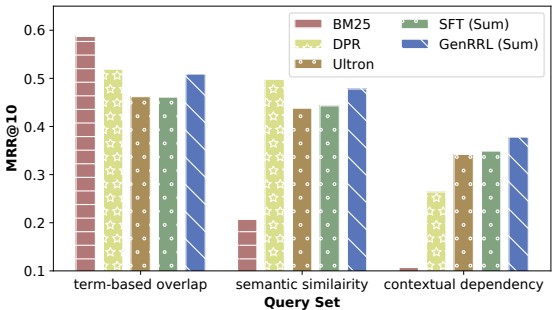

Figure 3: Performance on different query sets corresponding to three types of relevance scenarios on MS.

Table 3: The MRR@10 of different RL losses.

| Model | RL Loss | MS | NQ |
|-------|---------|-----|-----|
| GenRRL (URL) | Pointwise | 43.42 | 43.01 |
| GenRRL (URL) | Pairwise | 44.82 | 45.10 |
| GenRRL (URL) | Listwise | 45.93 | 45.73 |
| GenRRL (Sum) | Pointwise | 43.78 | 43.92 |
| GenRRL (Sum) | Pairwise | 45.37 | 45.46 |
| GenRRL (Sum) | Listwise | 46.62 | 46.31 |

provement on the search results. This progressive enhancement suggests that the listwise loss function could better alleviates the issue of the model's excessive focus on top-1 results. By assuming a non-deterministic distribution, the listwise loss assigns probability mass to other candidate docids based on their relevance.

## 6.4 Performance on Different Query Sets

To investigate the distinctions between the three relevance scenarios, we partitioned the test queries into three categories based on the performance of sparse, dense, and generative retrieval. These categories align with the dimensions of term-based overlap, semantic similarity, and contextual dependency, respectively. We then conducted a comparative analysis of the model's performance before and after reinforcement learning.

From Figure 3, we find that the BM25 and DPR models have demonstrated noticeable advantages on their respective query sets. Although the SFT model exhibited certain gaps in comparison, the introduction of reinforcement learning allowed us to incorporate relevance signals from both the sparse and dense models. Furthermore, the SFT model was augmented by incorporating the knowledge of the LLM. As a result, the retrieval quality across the three query sets experienced improvements. These enhancements signify the effectiveness of reinforcement learning process and diverse relevance signals, leading to enhanced retrieval performance.

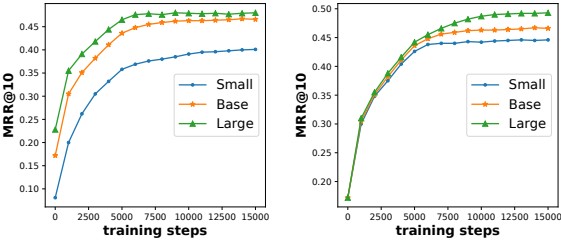

(a) Scale of SFT model.    (b) Scale of reward model.

Figure 4: The variation trends on MS with the increase of training step under different model scales.

## 6.5 Effect on Model Scale

To investigate the training characteristics of models with varying parameter scales, we try T5-small, T5-base and T5-large models for training SFT model and reward model respectively.

**Scale of SFT Model.** The learning curve depicted in Figure 4(a) illustrates the convergence behavior of the GenRRL model with varying parameter sizes as the number of training steps increases. It is observed that the small model exhibits comparatively lower MRR at convergence. Conversely, both the base and large models achieve similar performance levels after a sufficient number of training steps, with the large model slightly surpassing the base model. These findings emphasize the critical influence of model capacity on retrieval performance, indicating that larger models have the potential to yield improved results given adequate training.

**Scale of Reward Model.** Figure 4(b) shows the learning curve with different scale of the reward model. We observe that during the initial 4000 training steps, the differences between reward models of different sizes are not distinct. However, as training progresses, the small reward model exhibits early convergence. In contrast, the base and large reward models continue to improve their performance, with the large reward model outperforming the base model. This finding suggests that the larger reward model demonstrates stronger capability in capturing capture nuanced distinctions among sampled docids, leading to continued improvement.

## 7 Conclusion

In this paper, we address the challenges faced by generative retrieval models by proposing an RL-based approach called GenRRL. Our objective is to align the generative retrieval with the task of assessing document relevance, thereby improving

ranking performance. We introduce a three-stage reinforcement learning framework, where the relevance reward model captures different dimensions of relevance by considering term-based overlap, semantic similarity, and contextual dependency. Through extensive experiments, we demonstrate the effectiveness of our proposed method in improving generative retrieval performance.

## Acknowledgement

This work was supported by the National Natural Science Foundation of China No. 62272467 and No. 61832017, Beijing Outstanding Young Scientist Program No. BJJWZYJH012019100020098, Public Computing Cloud, Renmin University of China, Engineering Research Center of Next-Generation Intelligent Search and Recommendation, MOE, and Intelligent Social Governance Platform, Major Innovation & Planning Interdisciplinary Platform for the "Double-First Class" Initiative, Renmin University of China. The work was partially done at Beijing Key Laboratory of Big Data Management and Analysis Methods.

## Limitations

Despite the notable progress achieved within our reinforced framework, GenRRL encounters several challenges that necessitate further exploration and advancement. Primarily, the extension of the model to web scale poses substantial requirements for the design of the differentiable search index, demanding enhanced capacity to handle vast amounts of data. Secondly, the incorporation of new incoming documents into the differentiable search index has not yet been thoroughly investigated. This aspect presents an unexplored avenue that requires careful exploration and analysis. Addressing this challenge entails devising strategies and techniques to seamlessly integrate new documents into the existing model-based framework, while ensuring that the indexer retains its efficiency, accuracy, and robustness.

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
