# OpenReview forum: "Enhancing Generative Retrieval with Reinforcement Learning from Relevance Feedback"
_EMNLP/2023/Conference — EMNLP 2023 Main_

### Official Review · Reviewer_q2Hf · 2023-08-04

**Soundness:** 3

**Excitement:**

4: Strong: This paper deepens the understanding of some phenomenon or lowers the barriers to an existing research direction.

**Paper Topic And Main Contributions:**

The paper considers the problem of end-to-end generative retrieval, and targets two issues: 1) the discrepancy between token-level probabilistic optimization and the broader document-level relevance estimation; 2) an overemphasis on top-1 results. The authors then propose to solve these problems using Reinforcement Learning with Relevance Feedbacks.

**Questions For The Authors:**

1. Why the performance of DPR and ANCE are so low?
2. Can you elaborate more on the advantages of generative retrieval? The reasons from line 42 to 45 are not convincing to me. For example, why differentiable search index is advantageous compared to conventional explicit index? I think explicit index is more explainable, easier to include new documents.  Why end-to-end document generative model is prefer to, e.g. Dense Retrieval?

**Reasons To Accept:**

1. The paper is well motivated.
2. The paper shows that the proposed method is better than other generative retrieval baselines.
3. The writing is generally clear and easy to follow.

**Reasons To Reject:**

1. More explanation is needed to justify the reason for training the reward model.  Unlike RLHF in which the reward model is trained to mimic human preference which is unknown, here, we can directly use the combined scores of BM25, DPR or LLAMA as reward for Reinforcement Learning. I am wondering why do we need to train the reward model instead?
2. I have doubt on the experimental results as the performance of DPR and ANCE on Natural Questions are much lower than previously reported. For example, DPR can achieve Hit@1 of more than 40% (Table 5 in [1]), which is much higher than the reported result of ~23% reported in this paper. In addition, the author should cite the papers where the baseline results are reported if they are not reproduced by the authors.
3. The model is complicated, which requires the training of a dense model (DPR).

[1] Vladimir Karpukhin, Barlas Oguz, Sewon Min, Patrick Lewis, Ledell Wu, Sergey Edunov, Danqi Chen, and Wen-tau Yih. 2020. Dense Passage Retrieval for Open-Domain Question Answering. In Proceedings of the 2020 Conference on Empirical Methods in Natural Language Processing (EMNLP), pages 6769–6781, Online. Association for Computational Linguistics.

**Reproducibility:**

3: Could reproduce the results with some difficulty. The settings of parameters are underspecified or subjectively determined; the training/evaluation data are not widely available.

**Reviewer Confidence:**

3: Pretty sure, but there's a chance I missed something. Although I have a good feel for this area in general, I did not carefully check the paper's details, e.g., the math, experimental design, or novelty.

**Typos Grammar Style And Presentation Improvements:**

Updated: I would like to thank the authors for addressing my concerns, and updated the scores correspondingly.

---

> ### Author Rebuttal · Authors · 2023-08-29
>
> Thank you for your recognition of our motivation and empirical conclusions, and we are grateful for your thoughtful comments and inquiries regarding our work. In response to your inquiries, we offer the following clarifications.
>
> **Performance of DPR and ANCE**:
>
> In our study, both the reproduction of DPR and ANCE methods was based on their official codebase. The differences between our results and the original DPR paper primarily arise from two key factors:
>
> 1. **Dataset Disparity**: We employed the NQ320k dataset following [1], a dataset tailored for generative retrieval based on NQ, comprising 307k training samples and 7.8k test samples. Conversely, the original DPR paper used the Natural Questions dataset where documents were segmented into multiple passages, including 58k training samples and 3.6k test samples.
> 2. **Difference in Evaluation Methodology**: The DPR original paper considered retrieval results as correct if the retrieved documents contained the answer, allowing for multiple versions of wiki pages to potentially hold the correct answers. In contrast, our generative retrieval evaluation mandates the generated docid to be an exact match with the unique ground-truth document.
>
> These two factors collectively contribute to the observed result disparities. We apologize for any misunderstanding that may have arisen. All our experiments were conducted under identical settings to ensure the fairness and comparability of the results. Thank you for pointing out this issue; if given the opportunity, we will elucidate it further in the camera-ready version.
>
> [1] Yi Tay, et al. Transformer Memory as a Differentiable Search Index. in NeurIPS 2022
>
> **Advantages of Generative Retrieval**:
>
> Generative retrieval is an emerging approach to information retrieval that contrasts with traditional retrieval methods. While both aim to retrieve relevant information from a large corpus, they achieve this through fundamentally different mechanisms. Here are some advantages of generative retrieval over traditional retrieval methods:
>
> 1. **Reduce storage overhead**: Tradition index-based retrieval needs a large document index to search over the given corpus, which requires considerable memory resources to store all the data. In contrast, generative retrieval efficiently compresses extensive bodies of knowledge into the model's parameters, eliminating the need for large-scale disk storage of indexes. This mirrors the compactness and efficiency observed in the human brain, which can store vast amounts of information without taking up too much space.
>
> 2. **End-to-end optimization**: Traditional IR approaches are based on pipelined index-retrieval-rank framework, which is used in either dense and sparse retrieval models. Such a framework limits the optimization of the ranking results in an end-to-end way. End-to-end generative retrieval utilize a consolidated generative model, which is aware of the knowledge of all individual documents in the corpus, can be optimized end-to-end and increases the fitness of the model to the document retrieval task.
>
> 3. **Seamless Incorporation with LLMs**: As large language models evolve, they can be smoothly integrated into the generative retrieval framework, serving as its core. The exploration of generative retrieval offers a promising step towards advancing LLM for Information Retrieval (LLM4IR).
>
> 4. **Better Handling of Long-tail and Rare Queries**: Traditional dense retrieval methods might struggle with rare or long-tail queries since they rely on fixed vector representations. End-to-end generative retrieval models, due to their generative nature and powerful zero-shot capability of LM, can potentially generate relevant responses even for queries they haven't been explicitly trained on. This makes them more adaptive and capable of handling a wider range of search queries.
>
> In sum, while generative retrieval does have challenges, its unique advantages position it as a promising direction for future research in information retrieval. This paper focuses on aligning training objectives of generative retrieval with document relevance estimation, which combines the advantages of both generative and dense/sparse retrieval. Recently, some studies have focused on challenges such as incorporating new documents [1] and applying to larger corpora [2]. These efforts underscore the evolving nature of generative retrieval techniques. Moreover, we anticipate that the continuous advancement of LLM will play a pivotal role in further enriching the landscape of generative retrieval.
>
> [1] Sanket Vaibhav Mehta, et al. DSI++: Updating Transformer Memory with New Documents. Arxiv
>
> [2] Ronak Pradeep, et al. How Does Generative Retrieval Scale to Millions of Passages? Arxiv
>
> **Necessity of Reward Model**:
>
> Our initial strategy was indeed direct usage of combined scores from BM25, DPR, and LLAMA as rewards for reinforcement learning. However, when we weighted and summed the scores from the three annotators, we observed a decrease in performance. We think this was due to discrepancies in the scoring ranges and distributions across different model types, resulting in less-than-ideal reinforcement learning results. We did attempt normalization to address this, which yielded some improvements, but the gains remained minimal:
>
> | Reward Function              | Hit@1     | Hit@5     | Hit@10    | MRR@10    |
> | ---------------------------- | --------- | --------- | --------- | --------- |
> | Combined scores              | 29.72     | 59.93     | 68.02     | 42.52     |
> | Combined scores (normalized) | 30.82     | 61.46     | 70.28     | 43.27     |
> | Reward Model                 | **33.23** | **64.48** | **75.80** | **46.62** |
>
> To resolve this, we drew inspiration from RLHF's method of integrating multiple human annotator annotations. Instead of relying on raw scores, we asked each relevance annotator to produce a ranking. We then consolidated these rankings to generate pairwise labels to train our reward model. This approach effectively circumvented the issues from inconsistent scoring scales and biases across different models, providing a more comprehensive relevance feedback. More details regarding reward model can be referred to our response to the second reviewer (pWtF).
>
> **Model Complexity**:
>
> The primary contribution of this work lies in the proposal of a reinforcement learning framework tailored for generative retrieval. Under this framework, the complexity of model training can be adjusted according to specific requirements. For instance, in the annotator selection aspect, it allows for the integration of various existing models as black-box annotators (such as MonoBERT, RankT5), eliminating the need for training from scratch. During the model inference stage, invoking the docid generator alone is adequate, which only requires 0.04s of decoding top-10 docids for a query (comparable with sparse retrieval).
>
> ------
>
> We hope these clarifications address your concerns and provide a clearer perspective on our work's intentions and methodologies. We are grateful for your insights and feedback,which have been instrumental in refining our understanding and presentation of the research.
>
> Thank you once again for the time and effort you have invested in reviewing our manuscript and we are eager to address any additional concerns you may have. Your constructive feedback is invaluable in guiding the refinement of our research, and we look forward to possibly contributing a stronger and more refined piece to the conference.

---

### Official Review · Reviewer_pWtF · 2023-08-05

**Soundness:** 4

**Excitement:**

4: Strong: This paper deepens the understanding of some phenomenon or lowers the barriers to an existing research direction.

**Paper Topic And Main Contributions:**

This paper proposes a reinforcement learning-based approach to training differentiable search index (DSI). The authors first develop a relevance reward model, which is distilled from an ensemble of different relevance models. This reward model is then used to train DSI using reinforcement learning with three optimization strategies: pointwise, pairwise, and listwise strategies. Experiments on two datasets show that the proposed approach, GenRRL, outperforms the baseline methods.

The paper is overall interesting as it explores a novel direction to combine ranking losses in learning-to-rank with generative retrievers using reinforcement learning. The authors select to create the reward model by distilling from an ensemble of multiple off-the-shelf relevance annotators. I do have questions regarding some technical choices in the development of the rewarding model (listed below).

The experiments were conducted on subsets of MS MARCO Document Ranking and NQ. I find the improvement from SFT to GenRRL encouraging. The only concern is that there seem to be some differences between document representations of SFT/GenRRL and other baselines. Perhaps a fairer comparison would be to use the same document representation. However, I do not think this would be a reason to reject the paper.

A minor comment is to consider using a cross-attention ranker to replace DPR in the relevance annotators for reward model development, such as monoT5 [1] or RankT5 [2], which is shown to be effective in the RocketQA paper [3].

[1] Rodrigo Nogueira, Zhiying Jiang, Ronak Pradeep, and Jimmy Lin. 2020. Document Ranking with a Pretrained Sequence-to-Sequence Model. In EMNLP.

[2] Honglei Zhuang, Zhen Qin, Rolf Jagerman, Kai Hui, Ji Ma, Jing Lu, Jianmo Ni, Xuanhui Wang, and Michael Bendersky. 2023. RankT5: Fine-Tuning T5 for Text Ranking with Ranking Losses. In SIGIR.

[3] Yingqi Qu, Yuchen Ding, Jing Liu, Kai Liu, Ruiyang Ren, Wayne Xin Zhao, Daxiang Dong, Hua Wu, and Haifeng Wang. 2021. In NAACL.


**Questions For The Authors:**

Regarding the reward model development:

Question A: Is directly using the ranking metric like MRR or NDCG as the reward a bad idea? If so, why? Perhaps the authors could provide some experiments to show that this approach does not work well.

Question B: An alternative to Borda Rank is reciprocal rank fusion. Does this work as well as Borda Rank? The authors could provide some experiments to compare the two approaches.

Question C: In addition to pairwise loss, does pointwise/listwise distillation loss also work in *reward model training*? The authors could provide some experiments to compare the performance of these different loss functions.

I believe that providing this additional information would make the paper stronger and more convincing.

**Reasons To Accept:**

- Good observation on potential weakness of existing generative retrievers
- The paper studies different RL strategies to optimize the generative retriever
- Relatively thorough experiments on multiple data sets illustrating the effectiveness of RL



**Reasons To Reject:**


- A more fair comparison between baselines like DSI-QG etc. using the same doc_id strategy might better justify the effectiveness of RL
- Several steps in reward model training seem arbitrary and lack further explanations

**Reproducibility:**

4: Could mostly reproduce the results, but there may be some variation because of sample variance or minor variations in their interpretation of the protocol or method.

**Reviewer Confidence:**

2: Willing to defend my evaluation, but it is fairly likely that I missed some details, didn't understand some central points, or can't be sure about the novelty of the work.

**Typos Grammar Style And Presentation Improvements:**

Minor comments:
- Is there a missing negative sign in Line 340?
- Report the results of the 3 relevance annotators and their ensemble.

---

> ### Author Rebuttal · Authors · 2023-08-29
>
> Thank you for your meticulous review and appreciation of our technical originality. We genuinely value your insights and are grateful for the opportunity to address your questions. We hope that the following clarifications will allay your concerns and further consolidate our paper's position.
>
> ##  Explanations on the Reward Model:
>
> #### In Response to Quesion A:
> In our early experiments, we did consider ranking metrics such as MRR as a direct reward, drawing inspiration from [1] where Rouge served as the reward to enhance document summarization. However, the result on the MS MARCO dataset, as shown in the table below, shows the limitations of this approach:
>
> | Reward Function | Hit@1     | Hit@5     | Hit@10    | MRR@10    |
> | --------------- | --------- | --------- | --------- | --------- |
> | MRR             | 26.21     | 54.03     | 60.28     | 37.43     |
> | RM              | **33.23** | **64.48** | **75.80** | **46.62** |
>
> We've noticed that using MRR as the reward function resulted in a notable drop in performance. We deduced two primary reasons for this observed performance. Firstly, when the sampled docids does not include the ground-truth docid, the computation of MRR becomes unfeasible, thereby hindering effective model optimization. Secondly, employing MRR as the reward function presents limitations in that it only yields a single reward value for an entire list, failing to facilitate granular optimization of each token involved in the docid generation process.
>
> Therefore, based on these empirical observations and the identified limitations, we concluded that directly using ranking metrics like MRR may not be the most conducive to achieving optimal performance in generative retrieval tasks. To overcome these limitations, we decided to opt for a custom reward model (RM) that is capable of providing a more targeted and informative reward signal (relevance feedback). The robustness of this approach is substantiated by the improved performance metrics, as evidenced in our experiments.
>
> We hope these explanations clarify why we didn't use ranking metrics like MRR as direct rewards and that this will address any concerns you may have had on this aspect.
>
> [1] Yixin Liu, Pengfei Liu, Dragomir R. Radev, Graham Neubig. BRIO: Bringing Order to Abstractive Summarization. In ACL 2022.
>
> #### In Response to Question B:
>
> Reciprocal Rank Fusion and Borda Rank are both methods used in information retrieval and ranking scenarios. Upon your suggestion, we conducted experiments to compare these two algorithm. Here are the experimental results:
>
> | Rank Fusion            | Hit@1     | Hit@5     | Hit@10    | MRR@10    |
> | ---------------------- | --------- | --------- | --------- | --------- |
> | Reciprocal rank fusion | **33.36** | 64.45     | 75.61     | 46.54     |
> | Borda Rank             | 33.23     | **64.48** | **75.80** | **46.62** |
>
> The findings reveal comparable efficacy of both methods. While Reciprocal Rank Fusion exhibits a slightly higher score for the Hit@1 metric, Borda Rank outperforms or matches Reciprocal Rank Fusion in the Hit@5, Hit@10, and MRR@10 metrics. The results suggest that both methods are quite effective, and the choice between them may depend on the specific performance metrics that are of interest for a particular application. For instance, if the focus is on retrieving the most relevant document (Hit@1), Reciprocal Rank Fusion might be the preferred choice. On the other hand, if a balanced performance across different rank positions is desirable, Borda Rank appears to be more effective.
>
> We are genuinely grateful for the alternative suggestions you have presented. These suggestions offer valuable avenues for exploration and adaptation.
>
> #### For Question C:
>
> Our choice to initially utilize the pairwise loss was influenced by observations from InstructGPT and the approach delineated in "Learning to Summarize from Human Feedback." In addition to the pairwise loss, we also explored the use of pointwise and listwise distillation losses in reward model training. The challenge is that it becomes less straightforward to extract score distributions for individual documents (pointwise) or the entire list (listwise). Thus, to make it feasible to apply pointwise and listwise training schemes, we made a delineation: the topmost 25% of documents in the list were categorized as relevant and the remaining 75% as irrelevant. The results from our experiments comparing the different RM training losses are tabulated below:
>
> | RM Loss   | Hit@1     | Hit@5     | Hit@10    | MRR@10    |
> | --------- | --------- | --------- | --------- | --------- |
> | pointwise | 29.01     | 59.92     | 66.83     | 41.70     |
> | pairwise  | **33.23** | **64.48** | **75.80** | **46.62** |
> | listwise  | 31.26     | 62.01     | 70.68     | 43.89     |
>
> Upon examination, the pairwise loss  outperforms both the pointwise and listwise losses across all evaluated metrics. This merit may be due to its ability to capture nuanced differentiations between document pairs. In contrast, the pointwise and listwise methods exhibited relatively weaker performance, possibly because of biases from the binary relevant/irrelevant annotations. In light of the results, the pairwise loss emerges as the most effective strategy among the three in the context of our reward model training. Therefore, we adopted the pairwise loss in our model.
>
> We sincerely value your attention to these details and hope that our experiments and explanations provide clarity on our decision-making process and the outcomes achieved.
>
> ### Explanations On the Document Representations:
>
> In generative retrieval, mainstream document identifiers (docid) can be categorized into two major types: ID-based and keyword-based docids. We have chosen the keyword-based docid (comprising URL and summary) due to its capacity to leverage the strengths of pre-trained language models effectively and offer enhanced interpretability.
>
> Regarding your concerns on document representations, we further conducted experiments using semantic id as document representations. The table below shows our findings:
>
> | Model     | Document Rep. | Hit@1 | Hit@5 | Hit@10 | MRR@10 |
> | --------- | ------------- | ----- | ----- | ------ | ------ |
> | SFT (NCI) | semantic id   | 29.54 | 57.99 | 67.28  | 40.46  |
> | GenRRL    | semantic id   | 30.61 | 59.02 | 68.78  | 42.02  |
>
> The results demonstrate that our method consistently enhances model performance on ID-based docid. However, the degree of improvement is comparatively less pronounced than that achieved with keyword-based docids. This discrepancy might be attributed to the fact that keyword-based docids align more closely with the semantic space of the reward model. As a result, the feedback provided by the reward model can exert a more refined influence on the docid generator.
>
> Lastly, we'd like to acknowledge your suggestion concerning the inclusion of a cross-attention ranker in our model. This suggestion adds an intriguing layer to the conversation, and we plan to investigate the applicability and efficacy of a cross-attention ranker in future iterations of our project.
>
> We are hopeful that these explanations will fortify the paper's standing. Thank you again for your invaluable feedback and for helping us enhance the quality of our research.

---

### Official Review · Reviewer_UusY · 2023-08-06

**Soundness:** 4

**Excitement:**

4: Strong: This paper deepens the understanding of some phenomenon or lowers the barriers to an existing research direction.

**Paper Topic And Main Contributions:**

The work highlights the shortcomings of the recent generative retrieval models, namely overemphasis on the first document in the rank and lack of connection between the document-level relevance and token-level optimization, and suggest a novel reinforcement-based optimization to address these. The proposed GenRRL extends supervised training by defining a reinforcement loss function based an various reward definitions. The experiments on MS MARCO Document Ranking and Natural Questions datasets show that the method outperforms the recent generative retrieval models as well as some index-based dense retrieval ones.

**Reasons To Accept:**

- The paper is well-motivated, well-written. The preliminary experiments highlights the need for improving the optimization, and the method is clearly explained.
- The reinforcement-based method is novel and can be of interest of the community working in this area.
- The experiment results show the benefits of the method.

**Reasons To Reject:**

Nothing specific.

**Reproducibility:**

3: Could reproduce the results with some difficulty. The settings of parameters are underspecified or subjectively determined; the training/evaluation data are not widely available.

**Reviewer Confidence:**

2: Willing to defend my evaluation, but it is fairly likely that I missed some details, didn't understand some central points, or can't be sure about the novelty of the work.

---

> ### Author Rebuttal · Authors · 2023-08-29
>
> Thank you for the thorough and constructive review of our paper. We sincerely appreciate your recognition of the contributions of our work, particularly in identifying the novelty of our reinforcement-based optimization method, GenRRL, and its potential significance to the community.
>
> We are heartened by your positive feedback, which motivates us to maintain the standards of our work and possibly even to enhance it further in subsequent versions.

---

### Meta-Review · Area_Chair_HAJA · 2023-09-20

**Recommendation:** 5

**Metareview:**

This paper solves generative retrieval with a new reinforcement learning-based method, which is equipped with different reward models. Experiment results validate the effectiveness of the proposed method. The rebuttal has addressed most concerns of reviewers.

---

### Decision · Program_Chairs · 2023-10-07

**Decision:**

Accept-Main

**Comment:**

This paper solves generative retrieval with a new reinforcement learning-based method, which is equipped with different reward models. Experiment results validate the effectiveness of the proposed method. The rebuttal has addressed most concerns of reviewers.